# Promotion of homology-directed DNA repair by polyamines

Chih-Ying Lee[1], Guan-Chin Su [1], Wen-Yen Huang[2], Min-Yu Ko[1], Hsin-Yi Yeh[1], Geen-Dong Chang[1], Sung-Jan Lin [2,3,4,5] & Peter Chi [1,6]

Polyamines, often elevated in cancer cells, have been shown to promote cell growth and proliferation. Whether polyamines regulate other cell functions remains unclear. Here, we explore whether and how polyamines affect genome integrity. When DNA double-strand break (DSB) is induced in hair follicles by ionizing radiation, reduction of cellular polyamines augments dystrophic changes with delayed regeneration. Mechanistically, polyamines facilitate homologous recombination-mediated DSB repair without affecting repair via non-homologous DNA end-joining and single-strand DNA annealing. Biochemical reconstitution and functional analyses demonstrate that polyamines enhance the DNA strand exchange activity of RAD51 recombinase. The effect of polyamines on RAD51 stems from their ability to enhance the capture of homologous duplex DNA and synaptic complex formation by the RAD51-ssDNA nucleoprotein filament. Our work demonstrates a novel function of polyamines in the maintenance of genome integrity via homology-directed DNA repair.

[1] Institute of Biochemical Sciences, National Taiwan University, Taipei 10617, Taiwan. [2] Department of Biomedical Engineering, National Taiwan University, Taipei 10617, Taiwan. [3] Department of Dermatology, National Taiwan University Hospital, Taipei 100, Taiwan. [4] Department of Dermatology, College of Medicine, National Taiwan University, Taipei 10051, Taiwan. [5] Center for Developmental Biology and Regenerative Medicine, National Taiwan University, Taipei 10617, Taiwan. [6] Institute of Biological Chemistry, Academia Sinica, Taipei 11529, Taiwan. Correspondence and requests for materials should be addressed to P.C. (email: peterhchi@ntu.edu.tw)

Ornithine decarboxylase (ODC), the rate-limiting enzyme in the biosynthesis of polyamines (Supplementary Fig. 1a for chemical structures)[1], is upregulated in various types of human cancers[2,3]. 2-difluoromethylornithine (DFMO), a suicide inhibitor of ODC, suppresses cancer cell survival in mouse models and is being tested for its efficacy in cancer chemoprevention and the treatment of malignancies such as neuroblastoma[3,4]. Interestingly, DFMO has been found in a phase III trial to provide survival advantage when given as a post-radiation adjuvant[5]. Importantly, several studies have demonstrated that depletion of polyamines sensitizes cells to genotoxic substances, including ionizing radiation (IR), ultraviolet (UV), and etoposide[6–8], that can induce DNA double-strand breaks (DSBs). Taken together, the above findings point to an important, but as yet undefined role, of polyamines in the DNA damage response and maintenance of genome stability.

Here, we explore this. First, we reveal the importance of polyamines in DSB repair in a mouse hair follicle model. We then show that polyamines specifically upregulate homologous recombination (HR) but have little or no impact on non-homologous DNA end-joining (NHEJ) and single-strand DNA annealing (SSA). Mechanistically, polyamines promote RAD51-mediated DNA strand exchange by facilitating the capture of duplex DNA by the RAD51 presynaptic filament. Our study thus furnishes insights into the role of polyamines in genome maintenance via homology-directed DNA repair.

## Results

**Polyamines affect the DNA damage response in vivo.** The highly proliferative nature makes growing hair follicles highly susceptible to genotoxic injury, such as IR and chemotherapy[9]. We first tested the physiological relevance of polyamines in DSB repair in hair follicles after IR injury (see Fig. 1a for schematic). DFMO treatment or 2 Gy ionizing radiation had little or no impact on hair follicle growth. However, the combinatorial treatment with DFMO and radiation exerted profound dystrophy on hair follicle morphology (Fig. 1b). Specifically, after combinatorial treatment, the hair follicle length became much shortened with greater shrinkage of the hair bulbs (Fig. 1b and Supplementary Fig. 1b). In the hair bulbs, the number of highly proliferating matrix cells that fuel hair shaft elongation dwindled to ~50% of the normal level (Fig. 1c). Pulse BrdU incorporation and TUNEL assays further revealed that polyamine depletion decreased cell proliferation and increased apoptosis after IR injury (Fig. 1d and Supplementary Fig. 1c). Thus, polyamines help ensure the survival of proliferating cells upon the occurrence of DNA damage.

**Polyamine depletion impairs homology-directed DNA repair.** Next, we examined the effect of polyamine depletion on the elimination of DNA breakage induced by IR or by the poly ADP-ribose polymerase (PARP) inhibitor olaparib, which suppresses single-strand break repair and traps PARP onto DNA, thus leading to DSB formation. As assessed by the comet assay, DFMO-treated cells retained a much higher level of DNA damage over time (Supplementary Fig. 2a–d). Our results thus indicated that depletion of polyamines impairs the DSB repair capacity of cells but, since the initial level of DNA damage was the same in DFMO-treated and control cells based on the level of γH2AX (Supplementary Fig. 2e, f), it does not affect DNA lesion induction.

To further examine the involvement of polyamines in DSB repair, the DR-GFP, EJ5-GFP, and SA-GFP reporters were used to monitor the activities of the major DSB repair pathways, HR, NHEJ, and SSA[10–13], respectively (Fig. 2a and Supplementary Fig. 3a, d). Herein, HR, NHEJ, or SSA was triggered by expression of the endonuclease I-SceI, which generates a site-specific DSB in the reporter sequence[10–13]. In agreement with previous studies[7,14], treatment with DFMO significantly reduced levels of putrescine and spermidine but not that of spermine in the time point we analyzed (Fig. 2b and Supplementary Figs. 3b and 4a). Interestingly, such treatment attenuated HR activity in a dosage- and time-dependent manner without affecting NHEJ or SSA (Fig. 2c and Supplementary Figs. 3c, e and 4b). Analysis of the I-SceI protein level ruled out the possibility that HR deficiency could have stemmed from poor expression of the endonuclease in DFMO-treated cells (Fig. 2c and Supplementary Fig. 4b). Since HR is operational mostly in S/G2 phase cells, we asked whether polyamines affect cell cycle progression. However, profiling analysis showed that DFMO has only a minor effect on cell cycle progression (Supplementary Fig. 4c, d). It is worth noting that a prolonged S-phase in the DFMO treatment may be related to HR suppression. Consistent with the observation in U2OS cells, depletion of polyamines in HEK293 DR-GFP reporter cells also led to a significant attenuation of HR activity (Supplementary Fig. 4e, f). Importantly, attenuation of polyamines by DFMO treatment increased the sensitivity of HR-proficient cells (U2OS and wild-type DLD-1) to olaparib; but not HR-deficient cells including MDA-MB-436 (BRCA1 mutation) and BRCA2[-/-]-DLD-1 cells (Supplementary Fig. 4g). These results further strengthen our conclusion that polyamines promote homologous recombination-mediated DNA repair.

Next, we examined how suppression of *ODC* or overexpression of *SAT1* (spermidine/spermine N[1]-acetyltransferase), which catalyzes the export of polyamines[15], affects HR. Knockdown of *ODC* led to decreased levels of putrescine and spermidine, and overexpression of *SAT1* led to a significantly lower level of spermine (Fig. 2d). Importantly, HR activity became suppressed by either treatment (Fig. 2e), thus providing additional evidence for a function of polyamines in HR. Consistent with this notion, upregulation of cellular polyamines by overexpressing *ODC* significantly enhanced HR activity and corrected the HR deficiency caused by DFMO treatment (Fig. 2f, g).

**Polyamines influence RAD51 activity.** To further define the role of polyamines in HR, we first examined expression levels of key HR factors including RAD51, BRCA1, and BRCA2 but found that DFMO treatment has no effect on their cellular abundance (Supplementary Fig. 5a). We also asked whether the DNA damage-induced assembly of RAD51 nuclear foci would be dependent on polyamines. However, immunofluorescence staining revealed that the frequency of RAD51 foci is not influenced by polyamine depletion (Supplementary Fig. 5b). In summary, attenuating the synthesis of polyamines has no apparent impact on the expression of HR genes or the recruitment of RAD51 to DNA damage. Next, we investigated whether HR deficiency engendered by DFMO treatment might be overcome by an enhanced expression level of *RAD51*. The results indicated that a mild overexpression of *RAD51* by the use of the pCBA promoter led to a significant restoration of HR activity in DFMO-treated cells (Fig. 3a). This finding provides the first evidence that polyamines may influence the activity of RAD51 directly.

Next, we used the DNA strand exchange assay to examine the effect of polyamines on RAD51 recombinase activity (Fig. 3b). In the strand exchange reaction, the RAD51-ssDNA presynaptic filament engages a radiolabeled homologous double-stranded DNA (dsDNA) molecule, which is followed by exchange of strands of like polarity to yield a labeled DNA joint. We tested putrescine, spermidine, and spermine and found that they exerted a varying degree of stimulatory effect on DNA strand exchange

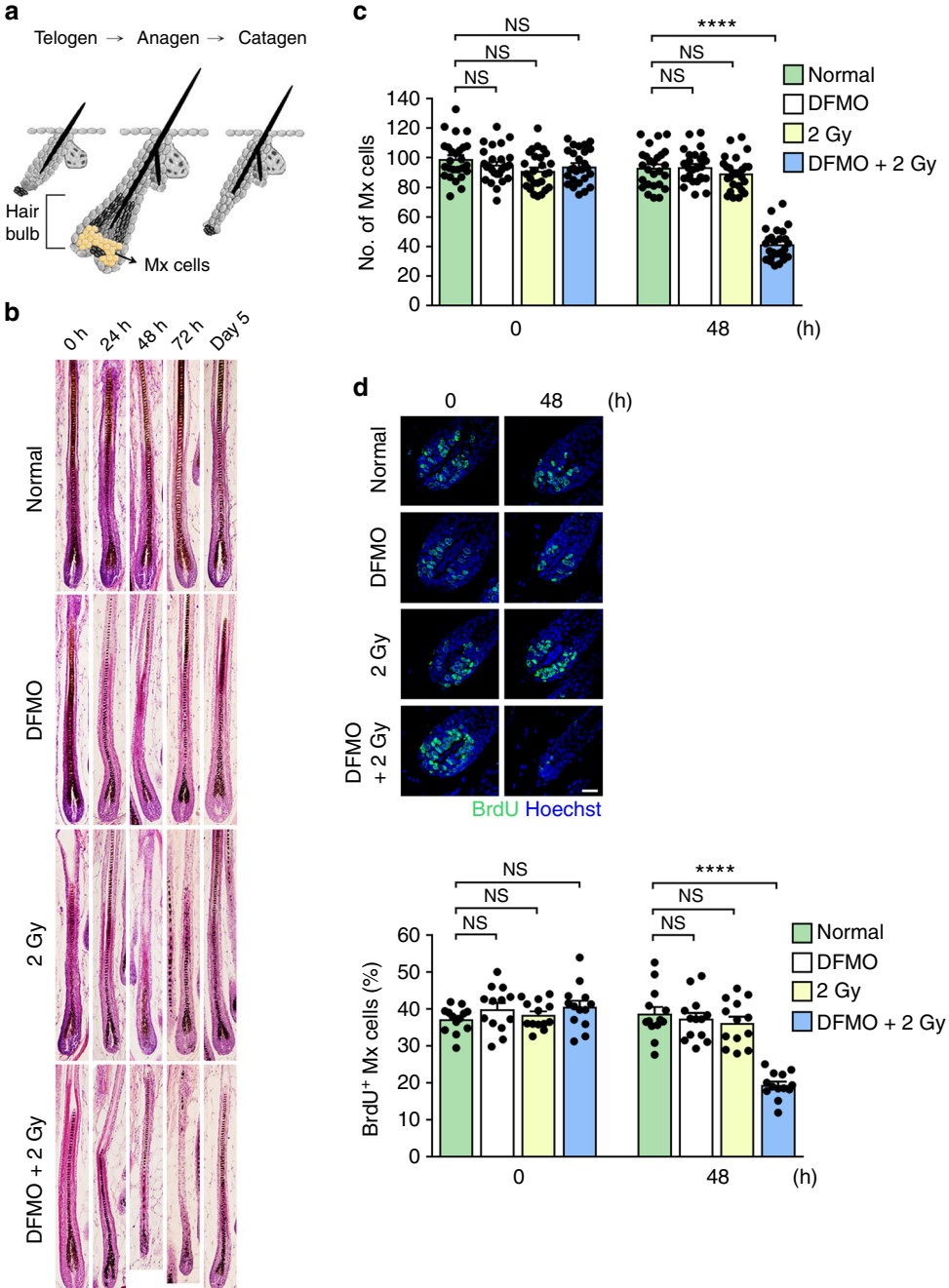

**Fig. 1** Protective role of polyamines in the DNA damage response in hair follicles. **a** The morphology of hair follicle and hair cycle. Hair cycle consists of three stages: hair growth (anagen), regression (catagen) and resting (telogen). The matrix cells (Mx) residing in the hair bulb actively proliferate during anagen to support the continuous elongation of hair shafts and to generate the internal structures of hair follicles. **b** Histology of murine hair follicles upon treatment with DFMO, IR, or both. Scale bar, 75 μm. **c** The number of matrix cells (Mx) after treatment with DFMO, IR, or both. (n = 26 hair follicles) **d** Cell proliferation as measured by pulse BrdU labeling. Representative images and percentages of BrdU positive matrix cells were shown. (n = 13 hair follicles). Scale bar, 25 μm. ****$P < 0.0001$; NS, not significant ($P > 0.05$). Data are the mean ± s.e.m. from three independent experiments. Statistics was performed by one-way ANOVA with Tukey's post hoc test. Source data are provided as a Source Data file

(Fig. 3c–f and Supplementary Fig. 6a–c). We also used the D-loop assay, which examines the incorporation of an oligonucleotide into supercoiled plasmid DNA that bears homology to the oligonucleotide, to reveal the stimulatory effect of polyamines on RAD51 activity (Supplementary Fig. 6d, e).

**Polyamines promote dsDNA capture by RAD51.** Mechanistically, the RAD51 stimulatory effect of polyamines could stem

from an enhancement of presynaptic filament assembly or stability, as has been documented for BRCA2 and SWI5-SFR1[16–18], or from an upregulation of the ability of the presynaptic filament to promote the assembly of the synaptic complex, as recently reported for the BRCA1-BARD1 complex[19]. We conducted relevant experiments to test these possibilities. First, protection of RAD51-bound ssDNA against digestion by *E. coli* exonuclease I was carried out to ask whether polyamines affect presynaptic filament assembly or stability (see schematic in Supplementary

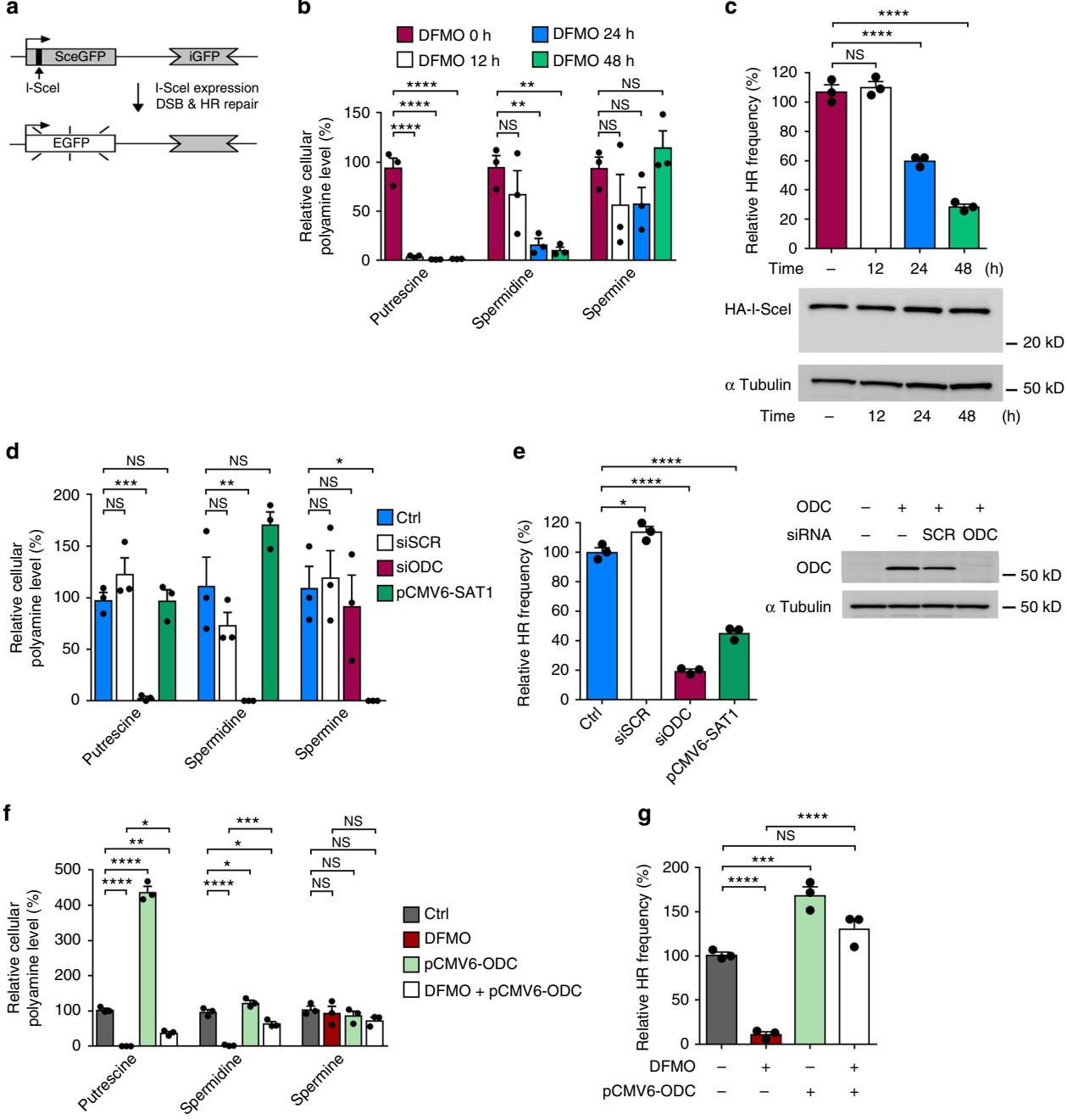

**Fig. 2** Polyamines affect DSB repair by homologous recombination. **a** Schematic of the DR-GFP reporter assay to assess HR activity. **b**, **c** U2OS cells were treated with 500 μM DFMO for the indicated times. **b** Levels of three polyamines were quantified and normalized to that in untreated cells. Note that, in agreement with previous studies[7,14], treatment with DFMO significantly attenuated levels of putrescine and spermidine but had a lesser effect on that of spermine. **c** Determination of HR efficiency using the DR-GFP reporter assay. GFP+ cells were quantified by flow cytometry 48 h after I-SceI transfection. Relative HR frequency was normalized to the percentage of untreated cells. Expression levels of I-SceI and tubulin were examined by immunoblotting. **d**, **e** U2OS cells were transfected without (Ctrl) or with scrambled siRNA (SCR), *ODC* siRNA, or expression vector of *SAT1* for 24 h. The cells were harvested for analyses 48 h after I-SceI transfection. **d** The level of individual polyamine was quantified and normalized to that in untransfected cells. **e** HR efficiency was determined using the DR-GFP reporter assay. The percentage of GFP+ cells was normalized to untransfected cells. Expression levels of ODC and tubulin were examined by immunoblotting. **f**, **g** The effects of DFMO (500 μM) treatment and *ODC* overexpression on HR efficiency were examined in U2OS cells. Untreated cells without any treatment or transfection were included as the control (Ctrl). **f** The relative level of individual polyamine was determined 2 days after overexpression of *ODC*. **g** HR efficiency was quantified by flow cytometry and normalized to control cells. *$P < 0.05$; **$P < 0.01$; ***$P < 0.001$; ****$P < 0.0001$; NS, not significant ($P > 0.05$). Data are the mean ± s.e.m. from three independent experiments ($n = 3$). Statistics was performed by one-way ANOVA with Tukey's post hoc test. Source data are provided as a Source Data file

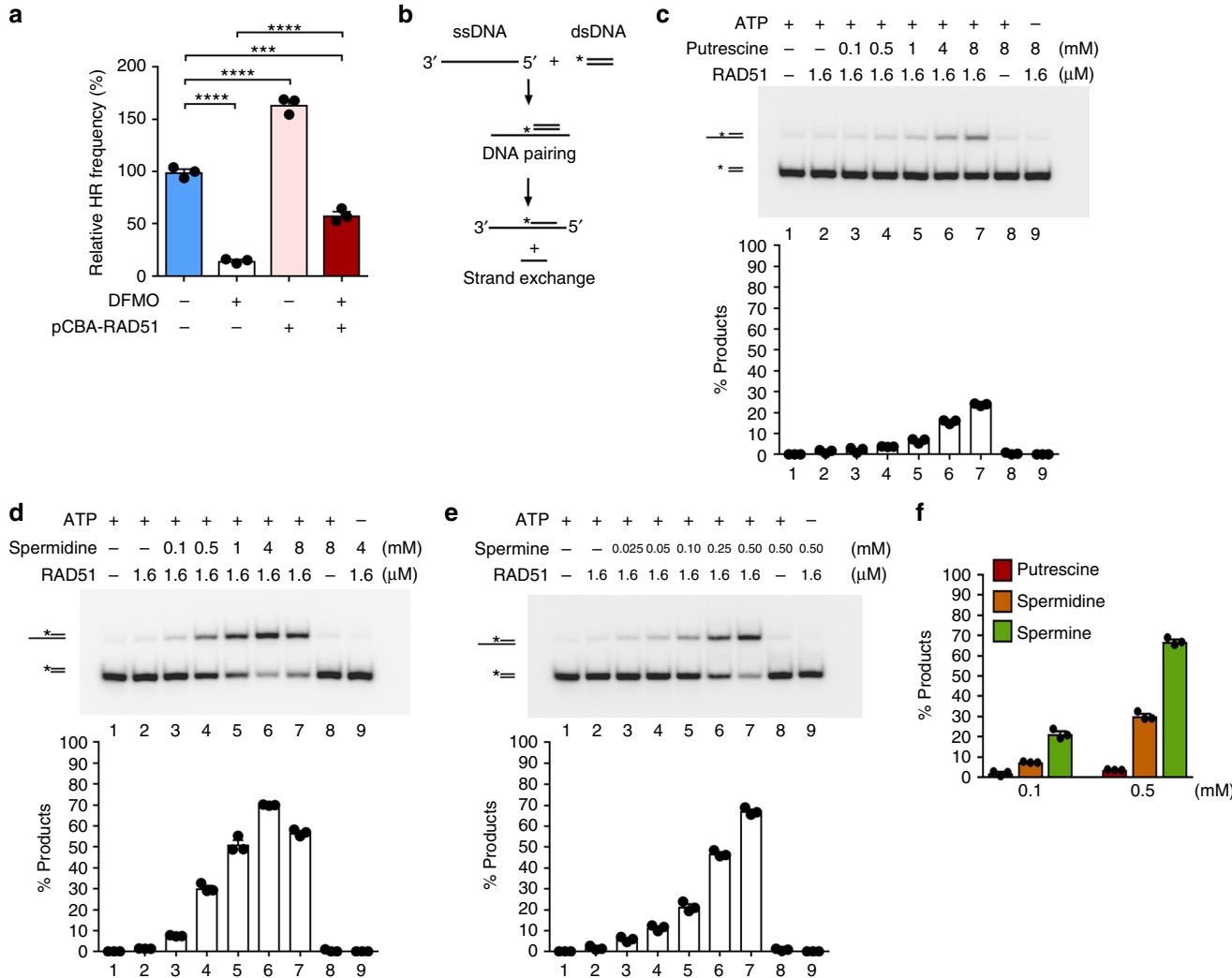

**Fig. 3** Polyamines stimulate RAD51-mediated DNA strand exchange. **a** The effects of DFMO (500 μM) treatment and *RAD51* overexpression on HR efficiency were examined in U2OS cells using the DR-GFP reporter assay. **b** Schematic of the DNA strand exchange assay. The asterisk denotes the [32]P-label. **c–e** Putrescine in **c**, Spermidine in **d**, or Spermine in **e** enhances the activity of RAD51-mediated DNA strand exchange in a concentration-dependent manner. Note that the ATP was required for strand exchange regardless of whether polyamine was present or not. The reaction time was 20 min. To normalize the percentage of product, each signal was subtracted from the percentage of product in control reaction (lane 1). **f** The effects of all three polyamines on RAD51-mediated DNA strand exchange were examined. The reaction time was 20 min. ***P < 0.001; ****P < 0.0001. Data are the mean ± s.e.m. from three independent experiments (n = 3). Statistics was performed by one-way ANOVA with Tukey's post hoc test. Source data are provided as a Source Data file

Fig. 7a). Consistent with cytological data showing normal assembly of DNA damage-induced RAD51 nuclear foci in DFMO-treated cells (Supplementary Fig. 5b), we found that polyamines showed little impact on DNA digestion by exonuclease I (Supplementary Fig. 7b, c). Thus, polyamines affect a step in RAD51-mediated DNA strand exchange after presynaptic filament assembly. Consistent with this idea, spermidine stimulated DNA strand exchange and D-loop formation even when the stability of the presynaptic filament was enhanced by the use of AMP-PNP as the nucleotide cofactor or by the inclusion of calcium ion to attenuate ATP hydrolysis by the presynaptic filament[20,21] (Supplementary Fig. 8).

We then examined the effect of polyamines on duplex DNA capture by the presynaptic filament. Here, we assembled RAD51 filaments on ssDNA immobilized on magnetic beads with AMP-PNP as the nucleotide cofactor. The presynaptic filaments were subsequently incubated with either homologous or heterologous duplex DNA and then followed by a wash with buffer. Then, the

resin was treated with SDS to elute RAD51 and associated dsDNA (see Fig. 4a for schematic). Importantly, we found that spermidine greatly enhanced duplex DNA capture (Fig. 4b), and that successful capture required a functional RAD51 filament as revealed by omitting the nucleotide cofactor or by the substitution of RAD51 with the RAD51 K133A mutant[21] that does not bind ATP (Supplementary Fig. 9a, b). We also confirmed that RAD51 K133A was incapable of DNA strand exchange even when spermidine was present (Supplementary Fig. 9c).

Upon the capture of dsDNA, the presynaptic filament conducts DNA homology search that leads to the formation of the synaptic complex in which the ss and dsDNA molecules are aligned in homologous registry with limited base pairing having occurred between the molecules[22]. The notion that polyamines may specifically enhance synaptic complex assembly is supported by our observation that a captured homologous dsDNA was retained more stably by the presynaptic filament than a non-homologous

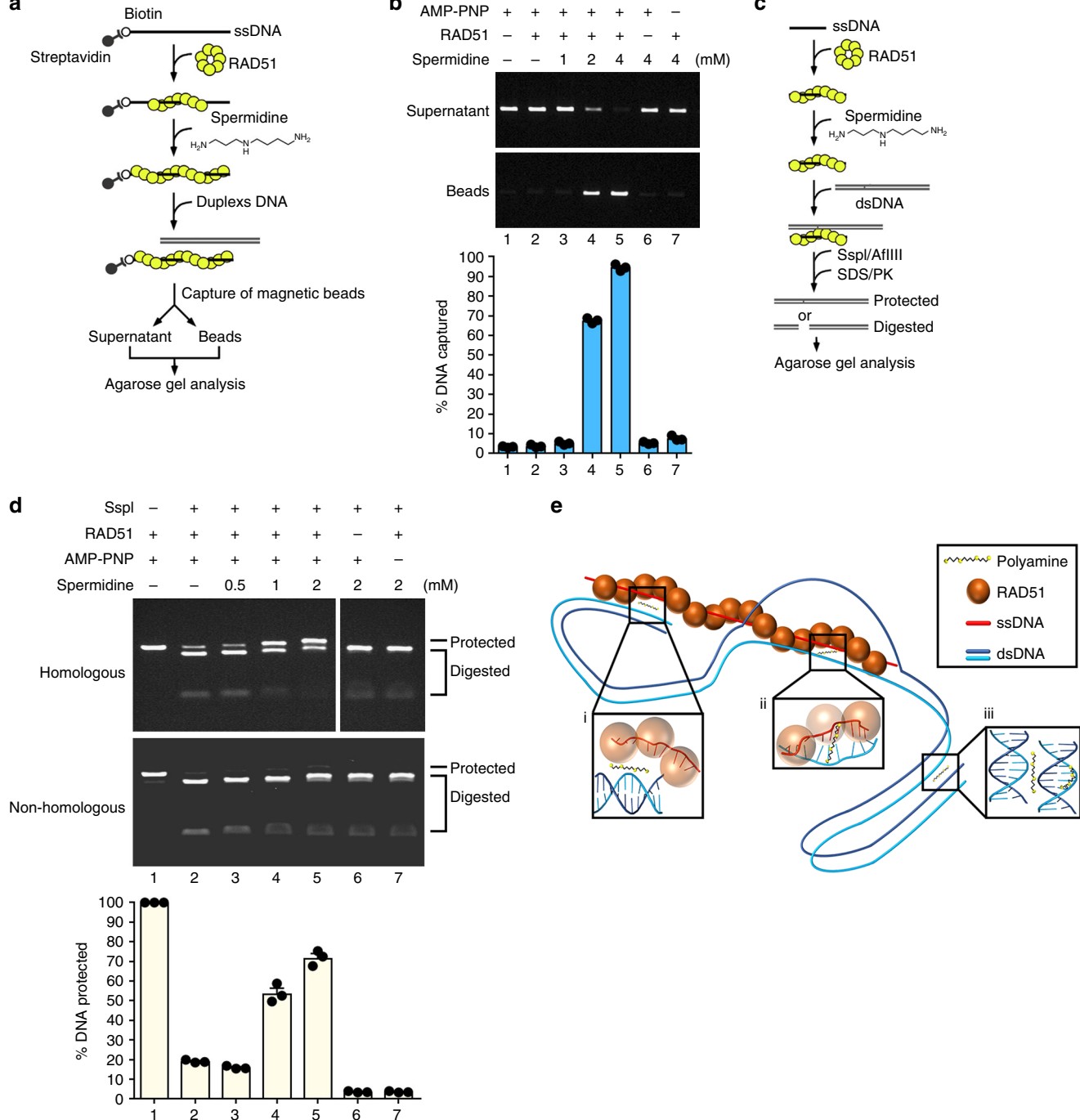

**Fig. 4** Polyamines enhance duplex capture and synaptic complex formation by RAD51. **a** Schematic of the duplex capture assay. **b** Duplex DNA capture by the RAD51 presynaptic filament was examined without or with spermidine. Note that AMP-PNP was used as the nucleotide cofactor to assemble a stable RAD51 nucleoprotein filament. **c** Schematic of the synaptic complex assay. Briefly, RAD51 presynaptic filament was incubated with duplex DNA containing either homology or non-homology of the restriction enzyme cutting site (SspI or AflIII) and then challenged with the indicated restriction enzyme (SspI herein or AflIII in Supplementary Fig. 9f). **d** Synaptic complex assembly by the RAD51 presynaptic filament was examined without or with spermidine. Following SspI digestion, the DNA species were resolved by agarose gel electrophoresis and stained with ethidium bromide. **e** Model depicting the mechanistic action of polyamines in promoting RAD51-mediated recombination. Three possible effects of polyamines in facilitating RAD51-mediated DNA strand exchange. i Polyamines promote the interaction between RAD51-ssDNA filament and duplex DNA. ii Polyamines could stabilize the invading ssDNA with the complementary ssDNA strand in the homologous duplex target during strand exchange. iii DNA condensation mediated by polyamines may facilitate intersegmental contact sampling[33] by RAD51 filament during homology search. Data are the mean ± s.e.m. from three independent experiments (n = 3). Source data are provided as a Source Data file

one (Supplementary Fig. 9d, e). Next, we used protection against restriction enzyme digest to quantify the effect of spermidine on synaptic complex formation[22,23] (Fig. 4c for schematic). The results revealed that spermidine enhanced synaptic complex formation markedly (Fig. 4d). The enhancement by spermidine was independently confirmed by using a different homologous pair (Supplementary Fig. 9f). As expected, enhancement of synaptic complex formation by spermidine required a functional presynaptic filament (Fig. 4d and Supplementary Fig. 9f, lane 7).

**Synthetic polyamine analogs lack activity**. Finally, we wished to determine whether the stimulatory effect of polyamines on RAD51-mediated reactions is specific. For this purpose, we tested the structural analog (carboxyethyl-γ-aminobutyric acid) and the ionic analog (*N*-(2-aminoethyl)−1,3-propanediamine) of spermidine and found that neither compound exhibited a significant effect on RAD51-mediated DNA strand exchange, D-loop formation, duplex DNA capture, or synaptic complex formation (Supplementary Fig. 10). These results highlighted the specificity of naturally occurring polyamines in RAD51 activity enhancement.

## Discussion

Much remains to be learned about how naturally occurring chemical compounds impact upon genome integrity. Several findings presented here and elsewhere support the conclusion that polyamines specifically promote HR in cells. First, the concentration of polyamines is elevated during the S and G2 phases of the cell cycle[24] wherein HR is most active[25]. Importantly, we have demonstrated that treatment of cells with DFMO to inhibit ODC or overexpression of *SAT1* to promote the export of polyamines lead to impaired HR. In contrast, *ODC* overexpression in DFMO-treated cells restored HR and lead to an enhanced efficiency of HR in untreated cells. Moreover, by using a mouse hair follicle model, we have shown that DFMO increased cell death in the proliferative cell compartment and delayed tissue regeneration after IR injury.

Our cell-based analyses indicated that all three polyamines contribute to HR. The range of physiological concentration of polyamines is about 0.88 to 1.58 mM in eukaryotic cells[26]. It is worth noting that the level of polyamines is often elevated in cancers, which is due to dysregulation of polyamine metabolism. For example, colorectal cancer harbors four times higher level of polyamines than the normal tissue[27]. Consistent with this premise, biochemical assays showed that all three polyamines, at physiologically relevant concentrations[26], stimulated RAD51-mediated homologous DNA paring and strand exchange activity to a varying degree. Mechanistically, we have furnished evidence that polyamines enhanced RAD51-mediated DNA strand exchange by promoting synaptic complex assembly. Since no direct interaction between RAD51 and polyamines could be detected (Supplementary Fig. 11), it seems unlikely that polyamines act by modifying the functional conformation of RAD51. Spermidine has been shown to stimulate yeast Rad51 recombinase activity[28]. Importantly, spermidine also stimulates RecA-mediated DNA strand exchange and D-loop formation in an ATP-dependent manner (Supplementary Fig. 12)[28]. Although we cannot rule out the possibility that the transient interaction of polyamines and RAD51 modulates protein conformation, our results favor that polyamines influence the properties of the DNA substrates to facilitate synaptic complex assembly rather than recombinase conformation. Altogether, these results provided evidence that polyamines have an evolutionarily conserved role in regulating recombinase activity.

It has been shown that polyamines can facilitate the pairing between two ssDNA strands and stabilize duplex DNA through interstrand interactions[29,30]. Conceivably, polyamines could enhance synaptic complex assembly by bridging ssDNA bound within the presynaptic filament to the incoming duplex DNA (Fig. 4e, inserted box i) and/or by stabilizing the nascent heteroduplex DNA joint formed as a result of DNA strand exchange (inserted box ii). Moreover, polyamines can function as bridging molecules between two DNA duplexes to increase DNA-DNA attraction[31]. Therefore, polyamines could promote the condensation of the incoming duplex DNA so as to increase the local concentration of DNA and increase the probability for the presynaptic filament to locate homology in the duplex partner (inserted box iii). These ideas are further supported by the correlation of spermine possessing the highest RAD51 stimulatory activity (Fig. 3f) and an optimal ability of dsDNA stabilization and condensation than putrescine or spermidine[29,32]

Based on work done with the *E. coli* RecA recombinase, it has been suggested that the three-dimensional structure of the incoming dsDNA is an important factor in the DNA homology search process[33]. Loop structures of dsDNA promote intersegmental contact sampling, which is mediated by RecA-ssDNA filament to exploit multiple weak contacts with dsDNA and facilitate efficient homology search. Consistent with this notion, volume-occupying agents, including polyethylene glycol (PEG), enhance RecA-mediated DNA strand exchange[34]. Thus, molecular crowding promotes recombinase-mediated homologous DNA pairing. As previously reported, polyamines possess the ability to condense DNA in vitro[35–37]. Thus, we propose that polyamines may have a role in intersegmental contact sampling through condensing dsDNA and producing transient loop structures. This idea is encapsulated in our working model presented in inserted box iii of Fig. 4e.

A recent study has provided evidence that the BRCA1-BARD1 tumor suppressor complex stimulates RAD51-mediated D-loop formation by promoting the assembly of the synaptic complex[19]. Our findings thus raise an intriguing question whether polyamines may act in parallel to or in synergy with BRCA1-BARD1 to facilitate RAD51-dependent DSB repair. Our findings also provide the impetus to evaluate the efficacy of combinatorial regimens involving the use of DFMO and olaparib or radiotherapy to treat tumors.

## Methods

**Cell culture and reagents**. Sources of cell lines were listed as following. U2OS cell lines that contain DR-GFP or EJ5-GFP reporter sequences were provided by Jeremy M. Stark. HEK293 cell line was provided by Matthew Porteus. MDA-MB-436 cell line was purchased from ATCC. The DLD-1 paired cell lines (parental and BRCA2$^{-/-}$) were purchased from Horizon Discovery. U2OS, MDA-MB-436, and HEK293 cells were cultured in Dulbecco's modified Eagle medium (DMEM, Gibco) supplemented with 10% fetal bovine serum (FBS, Biological Industries). DLD-1 paired cells were cultured in Roswell Park Memorial Institute (RPMI) 1640 medium (Gibco) supplemented with 10% FBS. All cells were cultured at 37 °C and 5% CO$_2$. One day before drug treatment or transfection of nucleic acids, the culture medium was replaced with DEME or RPMI 1640 supplemented with 10% dialyzed FBS (Gibco) to prevent polyamines in the culture medium from being transported into cells. All cell lines were tested negative for mycoplasma contamination by PlasmoTest (InvivoGen) and authenticated using short tandem repeat profiling. Difluoromethylornithine (DFMO) and olaparib were purchased from Tocris and Selleckchem, respectively. Putrescine, spermidine, spermine, carboxyethyl-γ-aminobutyric acid, 5-Bromo-2′-deoxyuridine (BrdU), and *N*-(2-Aminoethyl)−1,3-propanediamine were purchased from Sigma.

**Plasmids**. The human *ODC1* cDNA (RefSeq NM_002539.2) was cloned into *Not*I and *Pme*I sites of pCMV6-AC-GFP (OriGene). The human *SAT1* (RefSeq NM_002970.1) expression plasmid was purchased from OriGene. The human *RAD51* cDNA (RefSeq NM_002875.4) was cloned into the *Xba*I and *Pvu*II sites of pCBASceI to replace the *SCEI* coding region (Addgene). The SSA reporter plasmid, hprtSAGFP, was purchased from Addgene. Human RAD51 and RAD51 K133A

expression plasmids were constructed as described[21]. Human RPA expression plasmid was constructed as described[38] and kindly provided by Patrick Sung.

**Plasmid and siRNA transfection.** Plasmid transfection was carried out using Lipofectamine 3000 transfection reagent (Invitrogen). Briefly, cells were seeded in 60-mm dishes to be at 70% confluence on the day of transfection and the medium was changed 24 h post-transfection. For gene silencing, siRNAs targeting human *ODC1*, *MCPH1*, and non-targeting control were purchased from GE Dharmacon as ON-TARGETplus SMARTpool reagents. Briefly, cells were seeded in 60-mm dishes to be at 50% confluence on the day of transfection. Then siRNAs were transfected into cells using Lipofectamine RNAiMAX (Invitrogen) according to the manufacturer's instructions.

**Comet assay.** U2OS cells were seeded in 6-well plates at a density of $1 \times 10^4$ cells per well and treated with 500 μM DFMO for 72 h. Cells were washed with phosphate-buffered saline (PBS) twice and incubated with 125 nM olaparib for 24 h or irradiated with a 10 Gy ionizing radiation. Cells were then cultured with fresh medium containing 500 μM DFMO for the indicated recovery time. Comet assay (Trevigen) under alkaline condition was conducted according to the manufacturer's instructions. Briefly, cells were mixed with low-melting temperature agarose and spread onto slides. Slides were immersed in lysis buffer and then in unwinding solution for 1 h at 4 °C. Slides were then subjected to electrophoresis at 21 volts for 1 h at 4 °C, followed by staining with SafeView Plus dye (Applied Biological Materials) at room temperature for 40 min. Fifty randomly selected cells per sample were captured using a Leica DM6000B upright microscope system. The tail moment was computed as the percentage of DNA in the comet tail multiplied by the tail length in the individual nuclei using the CometScore software (TriTek).

**Cell cycle analysis.** Cells were washed with PBS, collected by centrifugation, and fixed in 70% ethanol at 4 °C for 24 h. The fixed cells were washed with PBS and stained in 500 μl PBS containing propidium iodide (20 μg ml$^{-1}$), ribonuclease A (200 μg ml$^{-1}$), and Triton X-100 (0.1% v/v) at 37 °C for 30 min in the dark. Cell cycle distribution of cells was calculated and analyzed using a FACSCalibur flow cytometer (BD Biosciences).

**Measurement of intracellular polyamines.** Sample preparation: Harvested cells with a number of $2 \times 10^6$ were added with 200 μl of ultrapure water and disrupted by sonication for 5 s, twice. Cell lysate was collected by centrifugation and 600 μl of acetonitrile was added to the cell lysate. The samples were vortexed rigorously for 1 min, and the particulate material was removed by centrifugation at $14,000 \times g$ for 10 min. Then, 450 μl of the supernatant was removed and vacuum-dried. The pellet was dissolved in 16 μl of ultrapure water, and 2 μl of 0.5 M carbonate buffer (pH 9.2) and 2 μl of 10 mg ml$^{-1}$ dansyl chloride prepared in acetone were added to the mixture. At the end of a 2-h incubation at 60 °C, 30 μl ultrapure water was added and the incubation continued for another 30 min. The samples were clarified by centrifuge at $14,000 \times g$ for 10 min, and 10 μl of the supernatant was subjected to liquid chromatography-electrospray ionization-tandem mass spectrometry (LC-ESI-MS) analysis.

LC-ESI-MS system: The LC-ESI-MS system consisted of an ultra-performance liquid chromatography (UPLC) system (Ultimate 3000 RSLC, Dionex) and an electrospray ionization (ESI) source of quadrupole time-of-flight (TOF) mass spectrometer (maXis HUR-QToF system, Bruker Daltonics). The autosampler was set at 4 °C. Separation was performed with reversed-phase liquid chromatography (RPLC) on a C18 column (2.1 × 100 mm, Walters). The elution started from 99% mobile phase A (0.1% formic acid in ultrapure water) and 1% mobile phase B (0.1% formic acid in acetonitrile). The following gradient was used: constant at 1% B for 0.5 min, from 1 to 60% B in 6 min, from 60 to 90% B in 0.5 min, constant at 90% B for 1.5 min, back to 1% B in 0.5 min. The column was equilibrated by pumping 1% B for 4 min. The flow rate was set at 0.4 ml min$^{-1}$ with an injection volume of 10 μl. LC-ESI-MS chromatogram was acquired under the following conditions: capillary voltage of 4500 V in positive ion mode, dry temperature at 190 °C, dry gas flow being maintained at 8 l min$^{-1}$, nebulizer gas at 1.4 bar, and an acquisition range of *m/z* 50–1000.

Data analysis: Data were processed by the TargetAnalysis and DataAnalysis softwares (Bruker Daltonics) with a summarized integrated area of signal for each metabolite. Compounds were identified according to their theoretical *m/z* with error tolerance of 20 ppm and mSigma value below 100 with the tolerance of LC peaks within a 0.2 min retention time shift and area higher than 200 counts. Levels of polyamines in cells were normalized to intracellular phenylalanine.

**Homologous recombination assay.** U2OS and HEK293 cell lines carrying the DR-GFP reporter were used in the analysis of HR activity as described[11]. Briefly, cells were seeded in 60-mm dishes to be 70% confluent at the time of transfection with the I-SceI expression vector pCBASceI (Addgene). Cells were then trypsinized and GFP-positive cells were sorted (Supplementary Fig. 13) in a FACSCalibur flow cytometer (BD Biosciences) at 48 h after pCBASceI transfection. To assess the effect of DFMO treatment on HR activity, cells were treated with DFMO (0.1 to 1 mM) for the indicated time prior to transfection with pCBASceI. For the silencing of *ODC1*, U2OS cells were transfected with control siRNA or siRNA specifically

targeting *ODC1* for 24 h. Cells were then cultured in fresh medium, followed by transfection with pCBASceI. For the overexpression of *ODC1* or *SAT1*, cells were transfected with pCMV6-ODC1 or pCMV6-XL5-SAT1 and incubated for 48 or 24 h, respectively, before the transfection with pCBASceI. For complementation studies, U2OS cells were pretreated with 500 μM DFMO for 24 h followed by overexpression of *ODC1* or *RAD51*. After a 48-h incubation, cells were cultured in fresh medium containing 500 μM DFMO followed by transfection with pCBASceI.

**Non-homologous end-joining assay.** U2OS cells carrying the EJ5-GFP reporter were used for the analysis of NHEJ activity as described[10]. U2OS cells were transfected with control siRNA or siRNA specifically targeting *MCPH1* for 48 h followed by transfection with pCBASceI. To study how depletion of polyamines might affect NHEJ, cells were treated with DFMO (0.1 to 2 mM) for 24 h prior to transfection with pCBASceI. After 72 h, cells were trypsinized and GFP-positive cells were sorted by flow cytometry as described above.

**Single-strand annealing assay.** HEK293 cells were transfected with SSA reporter plasmid, hprtSAGFP, for 48 h followed by treatment with DFMO (0.1 to 2 mM) for 24 h. Then, pCBASceI plasmid was transfected into cells. After 72 h, cells were trypsinized and GFP-positive cells were sorted by flow cytometry as described above.

**Antibodies.** The antibodies used were: anti-HA (Santa Cruz, sc-7392, for HA-I-SceI, 1:1000 dilution), anti-RAD51 (H-92, Santa Cruz, sc-8349, 1:1000 dilution for immunoblotting, 1:200 dilution for immunofluorescence), anti-BRCA1 (EMD Millipore, OP107, 1:1000 dilution), anti-BRCA2 (EMD Millipore, OP95, 1:500 dilution), anti-RAD51AP1 (Abcam, ab88370, 1:1000 dilution), anti-HOP2 (Proteintech, 11339–1-AP, 1:2000 dilution), anti-ODC1 (Abcam, ab66067, 1:4000 dilution), anti-γ-H2AX (clone JBW301, Millipore, 05–636, 1:1000 dilution for immunoblotting, 1:400 dilution for immunofluorescence), anti-H2AX (GeneTex, GTX 108272, 1:2000 dilution), anti-α tubulin (GeneTex, GTX112141, 1:10,000 dilution), and anti-BrdU (abcam, ab6326, 1:200 dilution). Horseradish peroxidase conjugated secondary antibodies were: anti-rabbit (GeneTex, GTX213110–01, 1:5000 dilution), anti-mouse (GeneTex, GTX213111–01, 1:5000 dilution). The secondary antibodies used for immunofluorescence were: DyLight 488 conjugated anti-rabbit (ThermoFisher, 35553, 1:200 dilution), DyLight 488 conjugated anti-mouse (ThermoFisher, 35502, 1:200 dilution). For hair follicle section staining, the secondary antibodies were purchased from Jackson ImmunoResearch as follows: Alexa Fluor® 488 Donkey Anti-Rat IgG (H + L) (712–545–150) and Cy™3 Donkey Anti-Rabbit IgG (H + L) (711–165–152).

**Immunoblotting analysis.** Cells were washed with PBS and incubated in lysis buffer (50 mM Tris-HCl, pH 7.5, 150 mM NaCl, 1% NP-40, 10% glycerol and 1 mM EDTA) containing 10 μg ml$^{-1}$ aprotinin, 10 μg ml$^{-1}$ chymostatin, 10 μg ml$^{-1}$ leupeptin, 10 μg ml$^{-1}$ pepstatin A, 1.5 mM PMSF, freshly prepared 2.5 mM sodium pyrophosphate, and 1 mM β-glycerolphosphate on ice for 20 min and then subjected to sonication. The lysate was clarified by centrifugation and protein concentration was determined using the BCA protein assay kit (Pierce). After sodium dodecyl sulphate polyacrylamide gel electrophoresis (SDS–PAGE), proteins were transferred onto polyvinylidene fluoride (PVDF) membrane. The membrane was treated with 5% bovine serum albumin (BSA) in PBS containing 0.01% Tween-20 (PBST) for 1 h and then incubated with the indicated primary antibodies at 4 °C overnight. After washing three times with PBST, the membrane was incubated with horseradish peroxidase (HRP) conjugated secondary antibodies at room temperature for 1 h. Then, the membrane was washed three times again with PBST before being incubated with enhanced chemiluminescent horseradish peroxidase substrate (ThermoFisher) for 5 min. Immunoblot images were acquired using the BioSpectrum imaging systems (UVP).

**Immunofluorescence staining.** U2OS cells were seeded in 24-well plates at a density of $5 \times 10^2$ cells per well and treated with 500 μM DFMO for 3 days before being exposed to 10 Gy ionizing radiation. Cells were fixed with 4% paraformaldehyde at 25 °C for 10 min and permeabilized in PBS containing 0.3% Triton X-100 for 10 min. After being washed with PBS twice, the cells were incubated in blocking buffer (5% BSA, 0.1% NP-40 in PBS) at room temperature for 30 min and with primary antibodies at 4 °C overnight. Next, the cells were washed with PBS containing 0.1% NP-40 and then incubated with fluorescently labeled secondary antibodies at 37 °C for 1 h. The cell nucleus was revealed by staining with 1 μg ml$^{-1}$ DAPI (4′,6-diamidino-2-phenylindole) at 25 °C for 2 min. Immunofluorescence images were acquired and analyzed in the IN Cell Analyzer system (GE Healthcare Life Sciences).

**Cell viability assay.** U2OS, MDA-MB-436, DLD-1 WT, and DLD-1 BRCA2$^{-/-}$ cells were seeded in 96-well plates at a density of $5 \times 10^2$, $2.5 \times 10^3$, $1 \times 10^2$, and $1.25 \times 10^3$ cells per well respectively. Cells were treated with the indicated amount of DFMO for 72 h. Cells were washed with PBS twice and incubated with the indicated amount of olaparib for 24 h before being returned to fresh medium for

72 h. Cell viability was determined using the alamarBlue cell viability reagent (ThermoFisher).

**DNA substrates**. All the oligonucleotides used were purified from a 10% polyacrylamide gel by electro-elution and filter-dialyzed in a Centricon-10 concentrator (Millipore) at 4 °C into TE buffer (10 mM Tris-HCl, pH 8.0, and 0.5 mM EDTA).

DNA substrates for DNA strand exchange: The 80-mer Oligo 1:

5′TTATGTTCATTTTTTATATCCTTTACTTTATTTTCTCTGTTTATTCAT TTAC
TTATTTTGTATTATCCTTATCTTATTTA was used for assembling the presynaptic filament. To prepare the target duplex 40-mer dsDNA, Oligo 2:
5′TAATACAAAATAAGTAAATGAATAAACAGAGAAAATAAAG was 5′ end labeled with polynucleotide kinase (New England Biolabs) and [γ-$^{32}$P] ATP (PerkinElmer). Following the removal of the unincorporated nucleotide with a Spin 6 column (Bio-Rad), the radiolabeled oligonucleotide was annealed to its exact complement Oligo 3, by heating to 85 °C for 10 min and slow cooling to 25 °C. The radiolabeled duplex substrate was purified from a 10% polyacrylamide gel.

DNA substrates for duplex DNA capture: The 5′ biotinylated 80-mer Oligo 1 or 90-mer Oligo 6 5′AAATCAATCTAAAGTATATATGAGTAAACTTGGTCTGA CAGTTACCAATGCTTAAT
CAGTGAGGCACCTATCTCAGCGATCTGTCTATTT was used for assembling the presynaptic filament. Note that pBluescript (Agilent), but not pRSFDuet (Novagen), plasmid DNA bears a region of homology (between positions 1932 and 2022) to Oligonucleotide 6.

DNA substrates for synaptic complex formation: The 60-mer Oligo 4: 5′AA TGTTGAATACTCATACTCTTCCTTTTTCAATATTATTGAAGCATTTATC AGGGTTATT is homologous to the region of pUC19 dsDNA (New England Biolabs) between positions 2471 and 2530 and that encompasses the target Ssp1 restriction enzyme site. The 60-mer Oligo 5:
5′CAGAATCAGGGGATAACGCAGGGAAAGAACATGTGAGCAAAAGGC CAGCAAAAGGCCAGGA is homologous to the region of pUC19 dsDNA between positions 779 to 838 and that covers the AflIII restriction enzyme site.

DNA substrates for D-loop assay: The 90-mer Oligo 6 was used for assembling the presynaptic filament and pBluescript plasmid DNA was used as the dsDNA substrate.

**Recombinant proteins**. Human RAD51 and RAD51 K133A were expressed in *E. coli* and purified as described[21]. Human RPA was expressed in *E. coli* and purified as described[39]. *E. coli* RecA and SSB proteins were purchased from New England Biolabs and Promega, respectively.

**DNA stand exchange assay**. All the reaction steps were carried out at 37 °C. The 80-mer Oligo 1 (4.8 μM nucleotides) was incubated with 1.6 μM RAD51 or RAD51 K133A in 10 μl of buffer A (35 mM Tris-HCl, pH 7.5, 1 mM dithiothreitol (DTT), 0.25 mM MgCl₂, 35 mM KCl and 80 ng μl⁻¹ BSA) containing 1 mM ATP or AMP-PNP for 5 min. For RecA-mediated DNA strand exchange reaction, 1.6 μM RecA was similarly incubated in buffer A containing 1 mM MgCl₂ and ATP for 5 min. The indicated polyamine or spermidine analog was added in 1.5 μl, followed by a 5-min incubation. The reaction was initiated by adding $^{32}$P-labeled 40-mer duplex (2.4 μM base pairs) in 1 μl. After a 20-min incubation, a 5 μl aliquot was removed and mixed with an equal volume of 0.1% SDS containing proteinase K (0.8 mg ml⁻¹) and incubated for 10 min. Samples were resolved in a 10% polyacrylamide gel in TBE buffer (89 mM Tris, 89 mM borate, and 2 mM EDTA, pH 8.0). The gel was dried onto DE81 paper (Whatman) and subjected to phosphorimaging analysis in a Personal FX phosphorimager using the Quantity One software (Bio-Rad).

**D-loop assay**. All the reaction steps were carried out at 37 °C. The $^{32}$P-labeled 90-mer Oligo 6 (2.4 μM nucleotides) was pre-incubated with 0.86 μM RAD51 in 10 μl buffer A containing 1 mM ATP or AMP-PNP for 5 min, followed by the addition of the indicated polyamine or analog in 1.5 μl and a 5-min incubation. For RecA-mediated reaction, 0.86 μM RecA was similarly incubated in buffer A containing 1 mM ATP and MgCl₂ for 5 min, followed by the addition of the indicated amount of spermidine in 1.5 μl and a 5-min incubation. The D-loop reaction was initiated by adding pBluescript dsDNA (36 μM base pairs) and 0.12 μM RPA or SSB in 2 μl. After a 5-min incubation, a 5 μl aliquot was removed and mixed with an equal volume of 0.1% SDS containing proteinase K (0.8 mg ml⁻¹) and incubated for 10 min. After electrophoresis in a 0.85% agarose gel in TBE buffer. The gel was dried onto DE81 paper and phosphorimaging analysis was carried out to visualize and quantify the radiolabeled DNA species.

**Exonuclease I protection assay**. RAD51 (1.3 μM) was incubated with the 5′-$^{32}$P-labeled 80-mer Oligo 1 (3 μM nucleotides) in 18 μl buffer A containing 0.1 mM ATP at 37 °C for 5 min. Following the incorporation of the indicated polyamine in 1 μl and a 5-min incubation, exonuclease I (1.5 units; New England Biolabs) was added in 1 μl. After 20 min of incubation, a 10 μl aliquot was removed and mixed with a 2.5 μl stop solution containing 240 mM EDTA, 0.2% SDS, and proteinase K (0.32 mg ml⁻¹) and incubated at 37 °C for 10 min. The samples were subjected to electrophoresis in a 10% polyacrylamide gel in TBE buffer, followed by gel drying and phosphorimaging analysis.

**Duplex DNA capture assay**. All the reaction steps were carried out at 37 °C. The 5′-biotinylated 80-mer Oligo 1 or 90-mer Oligo 6 was immobilized on streptavidin-coated magnetic beads (Roche) as described[21]. To assemble the presynaptic filament, 1.5 μl of magnetic beads containing either oligonucleotide (4.8 μM nucleotides) were incubated with 1.6 μM RAD51 or RAD51 K133A in 10 μl of buffer A containing 1 mM ATP or AMP-PNP for 5 min. Following the incorporation of spermidine or the indicated spermidine analog in 1.5 μl and a 5-min incubation, the reaction was initiated by adding linearized pBluescript plasmid DNA (which bears a region of homology to Oligo 6) or pRSFDuet dsDNA (which has no homology to either oligonucleotide) (4.8 μM base pairs each) in 1 μl and incubated for 10 min. The resin was captured using a magnetic device and the supernatant was set aside for analysis later. The captured resin was resuspended in 12.5 μl of buffer A containing 0.5% SDS and 1 mg ml⁻¹ proteinase K for 20 min to digest protein and to elute the captured dsDNA. The saved supernatant was also similarly treated with SDS and proteinase K. The deproteinized resin eluates and supernatant fractions were resolved in a 0.9% agarose gel in TBE buffer, and DNA was stained with ethidium bromide and quantified using the Image lab software (Bio-Rad). To investigate how tightly the captured dsDNA was held by the presynaptic filament, magnetic resin that contained the ensemble of presynaptic filaments and bound dsDNA was washed by 12.5 μl of buffer (25 mM Tris-HCl, pH 7.5, 10% Glycerol, 0.01% Igepal CA-630, 2 mM β-mercaptoethanol, 0.5 mM EDTA and 150 mM KCl) once before being subjected to the same analytical procedure.

**Synaptic complex assay**. All the reaction steps were carried out at 37 °C. The 60-mer Oligo 4 and Oligo 5 (9.6 μM nucleotides) that bear homology to the *SspI* site and the *AflIII* site of pUC19, respectively, were incubated with 3.2 μM RAD51 in 10 μl of buffer A containing 1 mM AMP-PNP for 5 min. The indicated amount of spermidine was added in 1.5 μl, followed by a 5-min incubation. Following the incorporation of linearized pUC19 dsDNA (12 μM base pairs) in 1 μl and a 5-min incubation, SspI (12.5 U) or AflIII (10 U) was added to the reaction, which was incubated for an additional 15 min. After deproteinization treatment with 12.5 μl of buffer A containing 0.5% SDS and 1 mg ml⁻¹ proteinase K for 20 min, reaction mixtures were resolved in a 0.9% agarose gel in TBE buffer. DNA species were stained with ethidium bromide and quantified using the Image lab software (Bio-Rad).

**Biotinylation of polyamines and RAD51 pulldown assay**. Each polyamine at 10 mM concentration was prepared in PBS, pH adjusted with HCl to pH 8.0. EZ-Link NHS-LC-Biotin (Thermo Scientific) at concentration of 20 mM was prepared in DMSO. The reaction containing 0.5 ml of each polyamine, 0.3 ml NHS-LC-biotin and 0.2 ml PBS was incubated for 1 h at room temperature. The reaction was stopped by the addition of 1 ml 10 mM ethanolamine in PBS and incubation for 1 h at room temperature.

To immobilize biotinylated polyamines on streptavidin-coated magnetic beads (Roche), 50 μl (10 mg ml⁻¹) magnetic beads were washed with 100 μl PBS for three times and then were incubated with 0.2 mM biotinylated polyamines (10 nmole) in 50 μl PBS for 1 h at 4 °C, followed by washing with 100 μl PBS for three times and resuspended in 50 μl PBS. After removing the supernatant, 10 μl magnetic beads containing biotinylated polyamines were incubated with 3 μg RAD51 in 20 μl reaction buffer (35 mM Tris-HCl, 1 mM DTT, 1 mM ATP, 0.25 mM MgCl₂, 35 mM KCl) for 20 min at 37 °C. The streptavidin-coated magnetic beads were captured and then treated with 20 μl of 2% SDS to elute any RAD51 that might have been retained on the beads. The supernatant and eluate, 9 μl each, were analyzed by 10% SDS–PAGE and Coomassie Blue staining.

**Animals**. All animal experiments were approved by the Institute for Animal Care and Use Committee at NTU. All animal experiments were performed with 4-week-old female C57BL/6 mice. The dorsal hairs of animals were carefully shaved without damaging the skin. The early full anagen of dorsal hair follicles was determined when the skin color changed to black, usually around postnatal day 30. Animals were randomly assigned to different groups for the following experiments ($n = 8$ in each group). A daily dose of DFMO (4 mg kg⁻¹) was administered intraperitoneally from postnatal day 30 to day 37. At postnatal day 32, animals were irradiated with a single dose of 2 Gy gamma irradiation from the dorsal side using $^{137}$Cs source-662 keV photons (dose rate 3.37 Gy min⁻¹, gamma irradiator IBL 637 from CIS bio International, France). BrdU was injected intraperitoneally 1 h prior to obtaining skin sample. Skin specimens were collected at the indicated time points following 2 Gy of radiation for the following analysis. To achieve the single-blind analysis, the data collection and analysis were performed by different laboratory members. Moreover, the data from four groups were only identified by a number for the downstream analysis.

Tissue preparation: Skin specimens were fixed in 4% paraformaldehyde at 4 °C for 12 h and paraffin-embedded after dehydration in alcohol concentration gradients. To analyze hair follicle morphology, the paraffin-embedded skin specimens were sectioned and stained with hematoxylin and eosin (H&E). Apoptotic cells were detected using the DeadEnd™ Fluorometric TUNEL System (Promega) according to the manufacturer's instructions. Immunofluorescence staining was performed with routine antigen retrieval as suggested by the antibody manufacturers. All fluorescent images were acquired using a confocal microscope

(SP5, Leica). To quantify BrdU+ and TUNEL+ cells, we acquired $1024 \times 1024$ pixels sequential scans with a 63x oil immersion objective lens (1.4 NA). Animals were maintained in the pathogen-free (SPF) facility of National Taiwan University (NTU).

**Statistics.** All statistical tests were performed using GraphPad Prism 7 (GraphPad Software) to analyze statistical significance. The normality of data was tested by D'Agostino-Pearson test or the Shapiro–Wilk test to confirm that the data were normally distributed. The sample variance was similar between the groups which was determined by the Brown-Forsythe test for multiple groups or F test for comparison between two groups. Multiple groups were compared using one-way ANOVA with Tukey's post hoc test. Unpaired two-tailed Student's $t$-test was utilized to compare between two groups. $P < 0.05$ was considered to be statistically significant. No statistical methods were used to predetermine sample size. No data was excluded from analysis.

**Reporting summary.** Further information on experimental design is available in the Nature Research Reporting Summary linked to this Article.

## Data availability

A Reporting Summary for this Article is available as a Supplementary Information file. The source data for all figures are provided as a Source Data file and are available from the corresponding author upon request.

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

## Acknowledgements

We thank J.M. Stark and M. Porteus for providing reporter cell lines. We thank the technical assistance of Technology Commons in College of Life Science and Center for Systems Biology, National Taiwan University with the measurement of intracellular polyamines. This work was supported by Academia Sinica (P.C.), National Taiwan University (P.C.), Taiwan National Health Research Institutes (NHRI-EX106–10410EI to S.J.L.), Taiwan Bio-Development Foundation (S.J.L.), and Taiwan Ministry of Science and Technology (MOST 105–2314-B-002–073-MY4 to P.C., MOST 105–2627-M-002–010 to S.J.L., and MOST 105–2311-B-002–013-MY3 to G.D.C.).

## Author contributions

C.Y.L. and P.C. conceived the study. C.Y.L., G.C.S., G.D.C., S.J.L. and P.C. designed the experiments. C.Y.L. and M.Y.K. performed cell-based experiments. G.C.S. and H.Y.Y. carried out biochemical studies. W.Y.H. performed animal analyses. C.Y.L. provided statistical analysis. C.Y.L. and P.C. wrote the paper. All authors discussed the results and contributed to the manuscript.

## Additional information

**Competing interests:** The authors declare no competing interests.

