## [Peer Review File · Nature Communications]

Reviewers' comments:

Reviewer #1 (Remarks to the Author):

The submitted manuscript by Lee, C-Y., et al entitled "Promotion of homology-directed DNA repair by polyamines" explores the ability of physiologically relevant polyamines in promoting DNA homologous recombination (HR) by RAD51. The authors provide convincing biochemical and cell based data in support of their conclusions that polyamines facilitate HR. Overall the study provides new and interesting insight into mechanisms of HR and is performed with high quality, and proper controls and statistical methods. The study also provides potential insight into how suppression of polyamines in cells can increase the efficacy of PARP inhibitors by suppressing HR activity.

Minor points to consider:

1. The study may have a wider impact if DFMO treatment or other drugs that reduce polyamines were shown to increase the efficacy of PARP inhibitors in BRCA1/2 mutated cell versus BRCA1/2 complemented cells.- such as breast cancer cell models (MDA-MB-436, HCC1937). If these assays are performed and successful then they should be highlighted in the abstract.
2. Line 66- is it true that PARP inhibitors cause replication-transcription collisions? If so, reference should be added.
3. Line 85- Perhaps DFMO shows some effect on cell cycle- slightly prolonged S-phase due to suppression of HR? RAD51 inactivation has been previously shown to cause S/G2 arrest. Therefore, one would expect some cell cycle arrest due to DFMO treatment.
4. It might be interesting to see if suppression of polyamine activity affects single-strand annealing reporter assay via RAD52.
5. Extended data Fig 7. Was ATP added to these protection assays? If not, they should be repeated with ATP or AMP-PNP since RAD51 is greatly impaired in ssDNA binding without ATP.
6. Line 172- would be informative to list physiological concentrations of known polyamines in cells, and whether these are altered in cancer for example.
7. Near line 130- it is important to discuss and reference previous studies on polyamine studies on condensing DNA in vitro and in cells. And how these previous findings correlate or conflict with current manuscript findings.
8. Around line 175- would be interesting to comment on/speculate whether polyamines are present in lower organisms i.e. bacteria and promote HR via RecA mechanisms- are the findings in the current manuscript conserved evolutionarily?

Reviewer #2 (Remarks to the Author):

The current manuscript provides evidence, using an impressive number of assays, that polyamines

have an important effect on Rad51-mediated homologous recombination. The overall effect is quite interesting, although not completely novel. There are a number of issues listed below that need to be addressed, without which the impact of this manuscript does not rise to the level warranting publication in Nature Communication.

1. There is the possibility that the sample size used in the animal experiments is not large enough to ensure reproducibility. How was it determined that only 6 animals were necessary if no pre-experiment statistics were employed. More importantly, why were so few hair follicles analyzed? And, is the number of hair follicles reported per animal? Or is that a total number? Further, all follicle analysis should be repeated blind, especially those from microscopy data

2. The authors failed to reference a previous study that showed that spermidine stimulates the Rad51 protein-mediated DNA strand exchange activity. See JBC (2001) v. 276 p. 38570 and add the reference, being sure to mention that this has been shown previously.

3. How did the authors determine that polyamines do not interact with Rad51 (line 175)? First, why is this negative result not shown, especially given that they shows so many other negative results? The problem here is that they rule out an effect on Rad51 conformation (line 177) and suggest that polyamines are affecting the DNA properties. This may very well be true. However, if this is the mechanism of Rad51 stimulation by polyamines, you would expect polyamines to also stimulate other recombinases, like RecA. This has been shown to not be the case. Again, see JBC (2001) v. 276 p. 38570.

4. The significance of the results shown in Figure 1 would be much more impactful if this effect were shown in human cell culture. The biochemical experiments were carried out using purified human proteins but the cellular effects were measured using a murine hair follicle model. There have been many examples of mouse model effects not translating to human.

5. Polyamines have been shown to decrease levels of reactive oxygen species in some species. Some of the effects shown in figure 1 may be attributable to reduced DNA damage as well as enhanced repair.

Reviewer #3 (Remarks to the Author):

It has been known that polyamines are elevated in cancer cells and promote cell growth. It has also been suggested that polyamines, specifically PEG, can facilitate strand exchange between DNA molecules in in vitro reactions through molecular crowding (most notably Lavery and Kowalczykowski JBC 267: 9307-9314, 1992). Here the authors examine the ability of polyamines to enhance the strand exchange activity of RAD51 recombinase. Whether this is related to the previous studies with PEG and RecA is not clear but those findings should be cited.

This issue notwithstanding, the authors show that there is an in vivo component to polyamines and DNA damage survival as they show that an inhibitor (DFMO) of ODC, a rate limiting enzyme for polyamine biosynthesis, sensitizes cells to DNA damage. After demonstrating the in vivo consequences of DFMO and suppression of homologous recombination (HR) activity, the authors focus on RAD51 activity and show that duplex DNA capture and D-loop formation is enhanced by polyamines.

The work has been very carefully executed and the manuscript contains a significant number of extended data figures. These studies show conclusively that polyamines enhance HR through action on

RAD51 activity, most likely through the DNA substrates. The discussion could have a section of molecular crowding and compare what was known about PEG and RecA activity in strand exchange to what is observed here for polyamines and RAD51 strand exchange.

Reviewer #1 (Remarks to the Author):

The submitted manuscript by Lee, C-Y., et al entitled “Promotion of homology-directed DNA repair by polyamines” explores the ability of physiologically relevant polyamines in promoting DNA homologous recombination (HR) by RAD51. The authors provide convincing biochemical and cell based data in support of their conclusions that polyamines facilitate HR. Overall the study provides new and interesting insight into mechanisms of HR and is performed with high quality, and proper controls and statistical methods. The study also provides potential insight into how suppression of polyamines in cells can increase the efficacy of PARP inhibitors by suppressing HR activity.

Minor points to consider:

1. The study may have a wider impact if DFMO treatment or other drugs that reduce polyamines were shown to increase the efficacy of PARP inhibitors in BRCA1/2 mutated cell versus BRCA1/2 complemented cells.- such as breast cancer cell models (MDA-MB-436, HCC1937). If these assays are performed and successful then they should be highlighted in the abstract.

(Author response) We thank the reviewer for the comment and suggestion. We now included more cell lines with a different BRCA1/2 genetic background to examine the efficacy of the PARP inhibitor olaparib in the condition with or without attenuation of the level of intracellular polyamines. Consistent with our findings, attenuation of polyamines by DFMO treatment increased the sensitivity of HR-proficient cells (U2OS and wild-type DLD1) to olaparib; but not HR-deficient cells including MDA-MB-436 (BRCA1 mutation) and BRCA2^{-/-}-DLD1 cells. These results further strengthen our conclusion that polyamines promote homologous recombination-mediated DNA repair. We have now included this information in the revised manuscript (Lines 105-109; Supplementary Fig. 4g). Note that we didn't highlight this in the abstract because we haven't done enough numbers of BRCA1/2 paired cell lines for this purpose. Thus, we feel that it is more appropriate to present this in a more conservative way.

2. Line 66- is it true that PARP inhibitors cause replication-transcription collisions? If so, reference should be added.

(Author response) Thank you for pointing this out. We have revised this sentence to “....ADP-ribose polymerase (PARP) inhibitor olaparib, which aborts single-strand break repair to lead to DSB formation.” (Lines 77-80).

3. Line 85- Perhaps DFMO shows some effect on cell cycle- slightly prolonged S-phase due to suppression of HR? RAD51 inactivation has been previously shown to cause S/G2 arrest. Therefore, one would expect some cell cycle arrest due to DFMO treatment.

(Author response) Thank you for pointing this out. Our original data do show a prolonged S-phase in a dosage- and time-dependent manner upon DFMO treatment. We now reformatted our figure presentation to a side-by-side comparison in each cell cycle phase in order for readers to easily discern this cell cycle effect (see Supplementary Fig. 4c, d). We also specifically discuss this point in the revised manuscript (Lines 100-103).

4. It might be interesting to see if suppression of polyamine activity affects single-strand annealing reporter assay via RAD52.

(Author response) We have utilized the single-strand annealing (SSA) reporter, SA-GFP, to study the role of polyamines in break repair by SSA. As presented in Supplementary Fig. 3e, depletion of polyamines has no significant effect on SSA activity. We have now included the above information (Lines 87-96; Supplementary Fig. 3e).

5. Extended data Fig 7. Was ATP added to these protection assays? If not, they should be repeated with ATP or AMP-PNP since RAD51 is greatly impaired in ssDNA binding without ATP.

(Author response) Indeed, we added ATP as the nucleotide cofactor in our reaction buffer (see Method, Line 468). To make it clear, we have added the label of ATP to Supplementary Fig. 7.

6. Line 172- would be informative to list physiological concentrations of known polyamines in cells, and whether these are altered in cancer for example.

(Author response) The range of physiological concentration of polyamines is about 0.88 mM to 1.58 mM in eukaryotic cells. It is worth noting that the level of polyamines is often elevated in cancers, which is due to dysregulation of polyamine metabolism. For example, colorectal cancer harbors 4 times higher level of polyamines than the normal tissue. We have now included the above information (Lines 201-205; References 26 and 27).

7. Near line 130- it is important to discuss and reference previous studies on polyamine studies on condensing DNA in vitro and in cells. And how these previous findings correlate or conflict with current manuscript findings.

(Author response) As reviewer's suggestion, we have now elaborated on this in the discussion (Lines 218-239).

8. Around line 175- would be interesting to comment on/speculate whether polyamines are present in lower organisms i.e. bacteria and promote HR via RecA mechanisms- are the findings in the current manuscript conserved evolutionarily?

(Author response) Polyamines indeed exist in all organisms including bacteria (Shah and Swiatlo, *Mol. Microbial.* **68**, 4-16, 2008). We found that spermidine also stimulated RecA-mediated strand exchange and D-loop formation. Thus, we think polyamines have an evolutionarily conserved role in regulating recombinase activity. Please also see our response to Point #3 of Reviewer 2. We have now included this information in the revised manuscript (Lines 212-217; Supplementary Fig. 12; Reference 28).

Reviewer #2 (Remarks to the Author):

The current manuscript provides evidence, using an impressive number of assays, that polyamines have an important effect on Rad51-mediated homologous recombination. The overall effect is quite interesting, although not completely novel. There are a number of issues listed below that need to be addressed, without which the impact of this manuscript does not rise to the level warranting publication in Nature Communication.

1. There is the possibility that the sample size used in the animal experiments is not large enough to ensure reproducibility. How was it determined that only 6 animals were necessary if no pre-experiment statistics were employed. More importantly, why were so few hair follicles analyzed? And, is the number of hair follicles reported per animal? Or is that a total number? Further, all follicle analysis should be repeated blind, especially those from microscopy data

(Author response) We thank the reviewer for the comment. No pre-experiment statistics were employed in our experimental design. However, the original data we presented could satisfy the requirement of minimum sample size (5 hair follicles) with type I and type II error as 0.01. To confirm the reproducibility of our findings, we have since repeated the animal study. We analyzed ten to thirty hair follicles additionally in a single-blind manner. Importantly, the data again reproducibly confirmed our observations that we had reported in the original manuscript. We have now included this information in the revised manuscript (Lines 530-533, Fig. 1 and Supplementary Fig. 1.).

2. The authors failed to reference a previous study that showed that spermidine stimulates the Rad51 protein-mediated DNA strand exchange activity. See JBC (2001) v. 276 p. 38570 and add the reference, being sure to mention that this has been shown previously.

(Author response) Thank you for pointing this out. We have amended the manuscript accordingly and added this to the revised manuscript (Lines 211-212; Reference 28).

3. How did the authors determine that polyamines do not interact with Rad51 (line 175)? First, why is this negative result not shown, especially given that they shows so many other negative results? The problem here is that they rule out an effect on Rad51 conformation (line 177) and suggest that polyamines are affecting the DNA properties. This may very well be true. However, if this is the mechanism of Rad51 stimulation by polyamines, you would expect polyamines to also stimulate other recombinases, like RecA. This has been shown to not be the case. Again, see JBC (2001) v. 276 p. 38570.

(Author response) Thank you for pointing this out. We utilized a pull-down assay to examine the interaction between RAD51 and different biotin-labeled polyamines (see detailed methodology in lines 504-518). Importantly, we found that none of polyamines interact with RAD51 directly (see Supplementary Fig. 11). As the reviewer suggested, we have now included the negative data in the revised manuscript.

Regarding possible stimulatory effect of polyamines on *E.coli* RecA, we have found that spermidine stimulates RecA-mediated strand exchange activity and D-loop formation in an ATP-dependent manner (see Supplementary Fig. 12). It is worth noting that Rice et al. pointed out that when the Mg^{2+} level is limiting (1 mM), spermidine has the positive effect on the production of nicked circular products by RecA-promoted DNA strand exchange with Φ X174 DNA substrates and nucleotide cofactor ATP (Rice et al., *J. Biol. Chem.* **276**, 38570-38581, 2001). Thus, we believe that polyamines have a major impact on the DNA properties rather than recombinase conformation. We have elaborated on this in the discussion and included the data in the revised manuscript. (Lines 209-217; Supplementary Fig. 12; reference 28).

4. The significance of the results shown in Figure 1 would be much more impactful if this effect were shown in human cell culture. The biochemical experiments were carried out using purified human proteins but the cellular effects were measured using a murine hair follicle model. There have been many examples of mouse model effects not translating to human.

(Author response) We thank the reviewer for making the point. We are aware that the observations from mouse model may not always translate to the human situation. Thus, we have expended considerable efforts by using human cell-based assays to demonstrate that polyamines play an important role in the regulation of DNA homology-directed double-strand break repair (see Fig. 2 and Supplementary Fig. 4). In addition, we have examined the combinatorial treatment with DFMO and PARP inhibitor olaparib in several human cell lines (see Supplementary Fig. 4g). Our results indicated that depletion of polyamines sensitizes HR-proficient cells to olaparib, which on its own would only induce cytotoxicity in HR-deficient cells. This finding provides evidence that polyamines play an important role in HR-mediated DNA repair in human cells. We have now included this information in the revised manuscript (Lines 105-109; Supplementary Fig. 4g).

5. Polyamines have been shown to decrease levels of reactive oxygen species in some species. Some of the effects shown in figure 1 may be attributable to reduced DNA damage as well as enhanced repair.

(Author response) We appreciate the reviewer's point that depletion of polyamines could elevate reactive oxygen species that would lead to increased DNA damage including double-strand breaks in mouse hair follicles. However, in our cell-based experiments, we have found that depletion of polyamines has limited, if any, effects on the level of double-strand breaks (see Supplementary Fig.

2).

Reviewer #3 (Remarks to the Author):

1. It has been known that polyamines are elevated in cancer cells and promote cell growth. It has also been suggested that polyamines, specifically PEG, can facilitate strand exchange between DNA molecules in in vitro reactions through molecular crowding (most notably Lavery and Kowalczykowski JBC 267: 9307-9314, 1992). Here the authors examine the ability of polyamines to enhance the strand exchange activity of RAD51 recombinase. Whether this is related to the previous studies with PEG and RecA is not clear but those findings should be cited.

(Author response) Thank you for the suggestion. Regarding RecA, we found that its strand-exchange activity can also be enhanced by spermidine (Lines 212-214, Supplementary Fig. 12). We have elaborated on this issue in the discussion and cited this reference in the revised manuscript (Lines 229-239, and reference 34).

2. This issue notwithstanding, the authors show that there is an in vivo component to polyamines and DNA damage survival as they show that an inhibitor (DFMO) of ODC, a rate limiting enzyme for polyamine biosynthesis, sensitizes cells to DNA damage. After demonstrating the in vivo consequences of DFMO and suppression of homologous recombination (HR) activity, the authors focus on RAD51 activity and show that duplex DNA capture and D-loop formation is enhanced by polyamines. The work has been very carefully executed and the manuscript contains a significant number of extended data figures. These studies show conclusively that polyamines enhance HR through action on RAD51 activity, most likely through the DNA substrates. The discussion could have a section of molecular crowding and compare what was known about PEG and RecA activity in strand exchange to what is observed here for polyamines and RAD51 strand exchange.

(Author response) In response to the Reviewer's point, we have now included a section in the discussion to elaborate on the effect of molecular crowding and the role of polyamines in DNA strand exchange mediated by RecA and RAD51 (Lines 229-239).

REVIEWERS' COMMENTS:

Reviewer #1 (Remarks to the Author):

The authors have satisfied all of my comments and suggestions. However, I suggest an additional change to the text lines 77-80 as follows:

"Next, we examined the effect of polyamine depletion on the elimination of DNA breakage induced by IR or by the poly ADP-ribose polymerase (PARP) inhibitor olaparib, which suppresses single-strand break repair and traps PARP onto DNA, thus leading to DSB formation."

Reviewer #2 (Remarks to the Author):

The revised manuscript addresses all of my previous concerns, except one. The pull-down showing that Rad51 does not interact with polyamines lacks a positive control, and may not show an interaction that is transient. Therefore, the discussion arguing that polyamines do not affect the conformation of Rad51 should be rewritten as one possibility.

Reviewer #3 (Remarks to the Author):

The authors have responded well to the reviewers' comments and have clarified issues and performed additional experiments. The paper provides an important study of the effect of polyamines on the HR reaction.

We are very grateful to the three reviewers for spending time to scrutinize our study and manuscript. As detailed below, we have addressed all the issues raised by the reviewers.

Reviewer #1 (Remarks to the Author):

The authors have satisfied all of my comments and suggestions. However, I suggest an additional change to the text lines 77-80 as follows:

“Next, we examined the effect of polyamine depletion on the elimination of DNA breakage induced by IR or by the poly ADP-ribose polymerase (PARP) inhibitor olaparib, which suppresses single-strand break repair and traps PARP onto DNA, thus leading to DSB formation.”

(Author response) We thank the reviewer for the comment. We have added this to the revised manuscript (Lines 78-80).

Reviewer #2 (Remarks to the Author):

The revised manuscript addresses all of my previous concerns, except one. The pull-down showing that Rad51 does not interact with polyamines lacks a positive control, and may not show an interaction that is transient. Therefore, the discussion arguing that polyamines do not affect the conformation of Rad51 should be rewritten as one possibility.

(Author response) Thank you for pointing this out. We have amended the manuscript accordingly and added this to the revised manuscript (Lines 207-210).

Reviewer #3 (Remarks to the Author):

The authors have responded well to the reviewers' comments and have clarified issues and performed additional experiments. The paper provides an important study of the effect of polyamines on the HR reaction.

(Author response) We appreciate the reviewer's point that our work provides an important study of polyamines on the homology-directed DNA repair.